# Has the introduction of direct oral anticoagulants (DOACs) in England increased emergency admissions for bleeding conditions? A longitudinal ecological study

Ana Alfirevic  ,[1] Jennifer Downing,[1] Konstantinos Daras,[2] Terence Comerford,[3] Munir Pirmohamed,[1] Ben Barr[4]

¹Department of Molecular and Clinical Pharmacology, University of Liverpool, Liverpool, UK
²Department of Geography and Planning, University of Liverpool School of Environmental Sciences, Liverpool, UK
³CLAHRC NWC Advisor Group, University of Liverpool, Liverpool, UK
⁴Department of Public Health and Policy, University of Liverpool, Liverpool, UK

**Correspondence to**
Professor Ana Alfirevic;
Ana.Alfirevic@liv.ac.uk

## ABSTRACT

**Objective** There is concern about long-term safety of direct oral coagulants (DOACs) in clinical practice. Our aim was to investigate whether the introduction of DOACs compared with vitamin-K antagonists in England was associated with a change in admissions for bleeding or thromboembolic complications.

**Setting** 5508 General practitioner (GP) practices in England between 2011 and 2016.

**Participants** All GP practices in England with a registered population size of greater than 1000 that had data for all 6 years.

**Main outcome measure** The rate of emergency admissions to hospital for bleeding or thromboembolism, per 100 000 population for each GP practice in England.

**Main exposure measure** The annual number of DOAC items prescribed for each GP practice population as a proportion of all anticoagulant items prescribed.

**Design** This longitudinal ecological study used panel regression models to investigate the association between trends in DOAC prescribing within GP practice populations and trends in emergency admission rates for bleeding and thromboembolic conditions, while controlling for confounders.

**Results** For each additional 10% of DOACs prescribed as a proportion of all anticoagulants, there was a 0.9% increase in bleeding complications (rate ratio 1.008 95% CI 1.003 to 1.013). The introduction of DOACs between 2011 and 2016 was associated with additional 4929 (95% CI 2489 to 7370) emergency admissions for bleeding complications. Increased DOAC prescribing was associated with a slight decline in admission for thromboembolic conditions.

**Conclusion** Our data show that the rapid increase in prescribing of DOACs after changes in National Institute for Health and Care Excellence guidelines in 2014 may have been associated with a higher rate of emergency admissions for bleeding conditions. These consequences need to be considered in assessing the benefits and costs of the widespread use of DOACs.

## INTRODUCTION

Prescribing of direct oral anticoagulants (DOACs), such as dabigatran, rivaroxaban,

### Strengths and limitations of this study

► The majority of general practitioner practices in England were included in our analyses and therefore we could estimate the effects across the whole population.
► Longitudinal data analyses included the changes in the guidelines for prescribing anticoagulants.
► We used effectiveness and safety data on anticoagulants used in England.
► It was not possible to link information on the prescribing for specific individuals to particular hospital admissions due to the ecological study design.
► No control for adherence to medications was possible.

apixaban and edoxaban that were licenced between 2013 and 2015, has increased rapidly in the UK. DOACs are used for stroke prevention in atrial fibrillation (AF), treatment of venous thromboembolism (VTE) and medical and surgical thromboprophylaxis.

Warfarin has been the primary oral anticoagulant in patients with AF and VTE for over 60 years. Its use is associated with large inter-individual variability in dose requirements, Narrow Therapeutic Index, slow onset of action, low time in the therapeutic range and concomitant drug and food interactions. In addition, regular international normalised ratio (INR) monitoring required for optimising loading and maintenance doses is burdensome to the patients and healthcare professionals. The DOACs are recommended by National Institute for Health and Care Excellence (NICE) guidelines for the treatment of adults with VTE, prevention of recurrent deep vein thrombosis, and stroke prevention in patients with non-valvular AF.[1] Several non-inferiority randomised

controlled trials have demonstrated that DOACs have comparable efficacy and safety profiles with warfarin.[2–5] In addition, in clinical trials, fixed-dose administration was proposed for DOACs, without the need for routine laboratory monitoring. This has led to the widespread uptake of DOACs into clinical practice. Although DOACs have been shown to be cost-effective in Randomised Control Trials (RCTs),[6] their high cost compared with the cost of warfarin (inclusive of INR testing) has major budget implications for the National Health Service (NHS) in the UK and other healthcare services globally.[7] Concern about the escalating costs to healthcare is increasing not only for DOACs, but also when complications occur, for their reversal agents such as idarucizumab or andexanet alfa.[8]

Data on the comparative safety and efficacy of DOACs in real-life observational studies have also become available more recently.[6] For instance, a recent large observational study in the UK found that apixaban was associated with the decreased risk of bleeding complications, but rivaroxaban and low-dose apixaban were associated with an increased risk of all-cause mortality compared with warfarin.[1] A large study (>14 500 patients) conducted in Scotland demonstrated that patients taking rivaroxaban may be at increased risk of bleeding compared with other DOACs.[9] Recent concern about bleeding risks associated with DOACs has prompted the European Medicines Agency (EMA) to launch a safety review (https://www.ema.europa.eu/documents/other/direct-oral-anticoagulants-doacs-ema-starts-review-study-bleeding-risk-direct-oral-anticoagulants_en.pdf).

In this study, we determined whether differences in prescribing of DOACs compared with warfarin between General Practitioner (GP) practices across England has been associated with variation of trends in emergency hospital admissions for bleeding and thromboembolic events from GP practice registered populations.

## METHODS
### Setting and data sources
We identified DOACs as dabigatran etexilate, apixaban, edoxaban or rivaroxaban using the British National Formulary (BNF) codes given in online supplementary appendix 1. We used the following NHS Digital data:

1. Practice level prescribing data that are published and made available by the NHS Business Service Authority each month. We used the Practice Prescribing Data File with the full 15-digit BNF code to calculate the annual DOAC prescribing rate as the percentage of DOAC items prescribed in proportion to all anticoagulants items prescribed for each GP practice in England between 2011 and 2016. The prescribing rate for each separate DOAC (dabigatran etexilate, apixaban, edoxaban or rivaroxaban) was calculated as the percentage of items of these medications prescribed as a proportion of all anticoagulants items prescribed. A prescription item refers to a single item prescribed on a prescription form, generally a course of medicine and is routinely used to measure trends in prescribing. We used the net ingredient costs of these items included in this data set to calculate the prescribing costs.

2. Hospital Episode Statistics data were used to calculate the annual emergency admission rate for bleeding and clotting conditions for each GP practice. The 10th revision of the International Statistical Classification of Diseases and Related Health Problems (ICD10) diagnostic codes included in this indicator is given in online supplementary appendix 1. Rates per 100 000 were calculated using annual data on the number of people registered with each GP practice provided by NHS Digital.

To additionally control for trends in the population age profile and underlying trends in morbidity, we also calculated the annual proportion of a GP population that was over 75 years and used annual data on the prevalence of AF, coronary heart disease (CHD) and chronic kidney disease (CKD) for each GP practice population using data reported through the Quality and Outcome Framework (QOF) returns.

We included all GP practices in England with a registered population size of greater than 1000 that had data for all 6 years. We excluded GP practices with values for prevalence estimates from QOF that were clear outliers (more than 2 IQRs below the first quartile or above the third quartile) as these probably represent reporting errors. In addition, we excluded practices where the practice registered population has changed by more than 20% between consecutive years as this would have reflected a large change in the practice population probably due to practice mergers or closures. This provided 5508 practices for the final analysis, each providing 6 years of data, that is 33 048 practice years of data. A flow chart is given in online supplementary appendix 2 detailing exclusions.

### Analyses
Initially, we investigated the geographical pattern of increases in DOACs prescribing across England. We mapped GP level prescribing data to lower super output areas (LSOAs), based on the proportion of each GP practice's population that lived in each LSOA. Next, we plotted national maps for the DOAC prescribing rate each year. We then used a fixed-effects Poisson regression model to investigate the association between the trend in prescribing within a GP practice and the trend in the rate of admissions for bleeding conditions. The outcome measure was the number of admissions and the log of the GP-registered population was used as an offset. As there is potential confounding from unobserved factors that vary between GP practice populations, we used a fixed-effects approach to remove these between-GP practice differences.[10] This conservative approach is the equivalent to including dummy variables for each GP practice so that the model assesses the association between the trend in prescribing and the trend in hospital admissions within each GP practice. We additionally included a dummy variable for each year to account

for the national trend in prescribing and emergency admissions. To additionally control for differential trends in need for anticoagulants and risk of bleeding complications in each GP population, we included annual measures of the prevalence of AF, CHD and CKD and the proportion of the GP population over 75-year-olds. We used a generalised estimating equation to account for the clustering of variance between GP practices.[11] To investigate whether there were different effects associated with different types of DOACs, we repeated the analysis for dabigatran, apixaban and rivaroxaban. As edoxaban was only licenced in the middle of 2015, there was insufficient data to analyse this separately. We present the effect estimates (rate ratios (RRs)) per additional 10% of DOACs prescribed, rather than for each additional 1 percentage point increase as this reflects a more meaningful level of increase than a 1 percentage point increase. This is achieved by dividing the RR by 10 to give the effect per 10 percentage points of DOAC prescribing.

### Sensitivity analysis

We subjected our analysis to a number of tests to assess the robustness of our findings. We estimated a negative binomial model as opposed to a Poisson model which is more robust to over-dispersion in the data, we also estimated a model using Huber-White clustered SE instead of a generalised estimating equations[12] and a model with autoregressive correlation structures, and replicated our models using data from all GP practices (ie, not applying any exclusion). To test the specificity of results, we estimated our model using a gastrointestinal infection as non-equivalent dependent variables.[13] This outcome should not be influenced by a change in the exposure but could be influenced by unobserved confounding factors that influence general trends in hospitalisation in a GP practice population.

### Patient involvement

No patients were involved in setting the research question or the outcome measures, nor were they involved in developing plans for design or implementation of the study. Patients and patient representatives within the National Institute for Health Research, Collaboration for Leadership in Applied Health Research and Care North West Coast (NIHR CLAHRC NWC) were asked to revise the manuscript, in particular, the lay language summary. The results of the study are going to be disseminated to the relevant patient and public groups through the CLAHRC NWC communication programme.

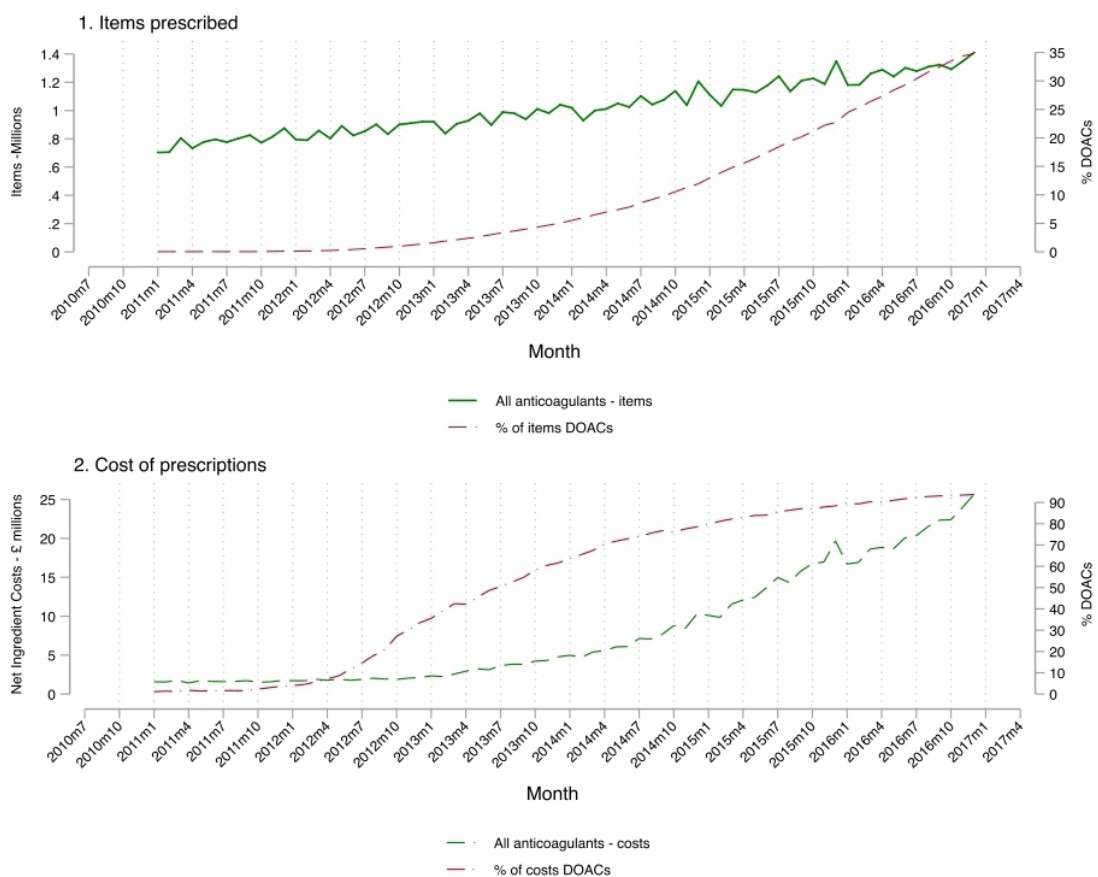

**Figure 1** Trend in anticoagulant prescribing items and costs from general practitioner practices in England and the proportion of direct oral anticoagulants (DOACs) of all anticoagulants prescribed. In 2014, National Institute for Health and Care Excellence guidelines on the use of DOACs has changed.

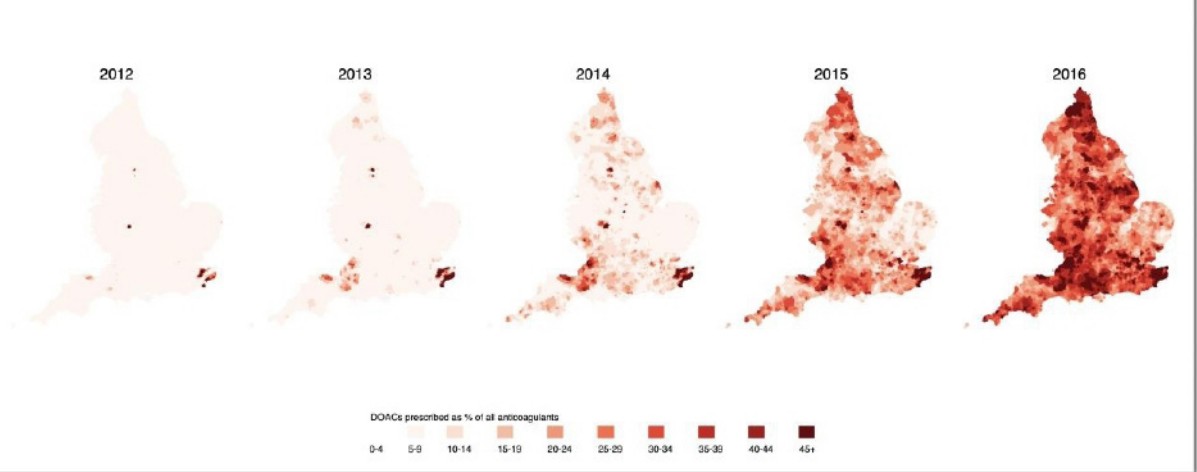

**Figure 2** Geographical pattern in direct oral anticoagulants (DOACs) prescribing as a proportion of all anticoagulant prescribing items in England 2011–2016 that includes changes in National Institute for Health and Care Excellence guidelines in 2014.

## RESULTS

Figure 1 shows the trend in the number of items and net drug costs of DOACs prescribed in England by GP practices between 2011 and 2016. Between 2011 and 2016, the number of items of anticoagulants prescribed had risen from 0.7 million to 1.4 million per month and the proportion of these that were DOACs had risen from <1% to 35%. Costs of anticoagulants rose more steeply, from £1.6 million per month in 2011 to £26 million per month by the end of 2016. This rise in costs was due to increased prescribing of DOACs, which by the end of 2016 accounted for 94% of the cost of all anticoagulant prescribing.

Figure 2 shows the geographical pattern of the DOAC prescribing each year. The increase in uptake of DOACs has not been consistent across the country, with the prescribing rate increasing markedly more in some part of the country compared with others. In particular, the areas around Somerset, Kent and to a lesser extent the Northeast appeared to have adopted DOACs sooner than other areas. By 2016, there was markedly lower usage of DOACs in the Northwest and East Anglia.

The results of the regression model indicate that differences in the trends in DOAC prescribing between GP practices from 2011 to 2016 were associated with differences in the trends in admissions for bleeding conditions. For each additional 10% of DOACs prescribed as a proportion of all anticoagulants, there was a 0.8% increase in emergency admissions for bleeding complications (RR: 1.008 95% CI 1.004 to 1.013). We present the effect estimates per additional 10% of DOACs prescribed as this reflects a more meaningful level of increase than a 1 percentage point increase. In other words, for each additional 1% of DOACs prescribed as a proportion of all anticoagulants, there was a 0.08% increase in emergency admissions for bleeding complications (RR: 1.0008 95% CI 1.0003 to 1.0013). To give the effect per 10% of DOACs prescribed as presented above, we have just divided the results for 1% change by 10. The increased risk of emergency admissions for bleeding complications was particularly associated with the increased use of dabigatran. For each additional 10% of DOACs prescribed as a proportion of all anticoagulants, there was a 0.5% reduction in emergency admissions for clotting complications (RR: 0.995 95% CI 0.99 to 1.0004), p=0.08; see figure 3. The CIs suggest that this result could have occurred even if there was no true association between DOAC prescribing and emergency admissions for clotting complications.

Figure 4 shows the trend in emergency admissions for bleeding conditions between 2011 and 2016 and the trend, estimated from the regression model that would have been expected if DOACs had not been introduced. Overall the introduction of DOACs between 2011 and 2016 was associated with 1.94 additional emergency admissions for bleeding complications per 100 000 people per year (0.98–2.9), the equivalent to a total of 4929 emergency admissions for bleeding complications (95% CI 2489 to 7370).

Our results were similar when using alternative model specifications (see online supplementary appendix 3). We found similar results when using a negative binomial model as opposed to a Poisson regression, when using Huber-White SEs or generalised estimating equations with alternative correlation structures and when replicating our model using data from all GP practices (ie, not applying any exclusions). We found no association between the trend in DOAC prescribing and emergency admissions when applying the model with a non-equivalent dependant variable (gastrointestinal infections).

## DISCUSSION

In this study, we demonstrated a sharp increase in the uptake of DOACs by GP in the UK between 2011 and 2016, compared with the current standard, warfarin.

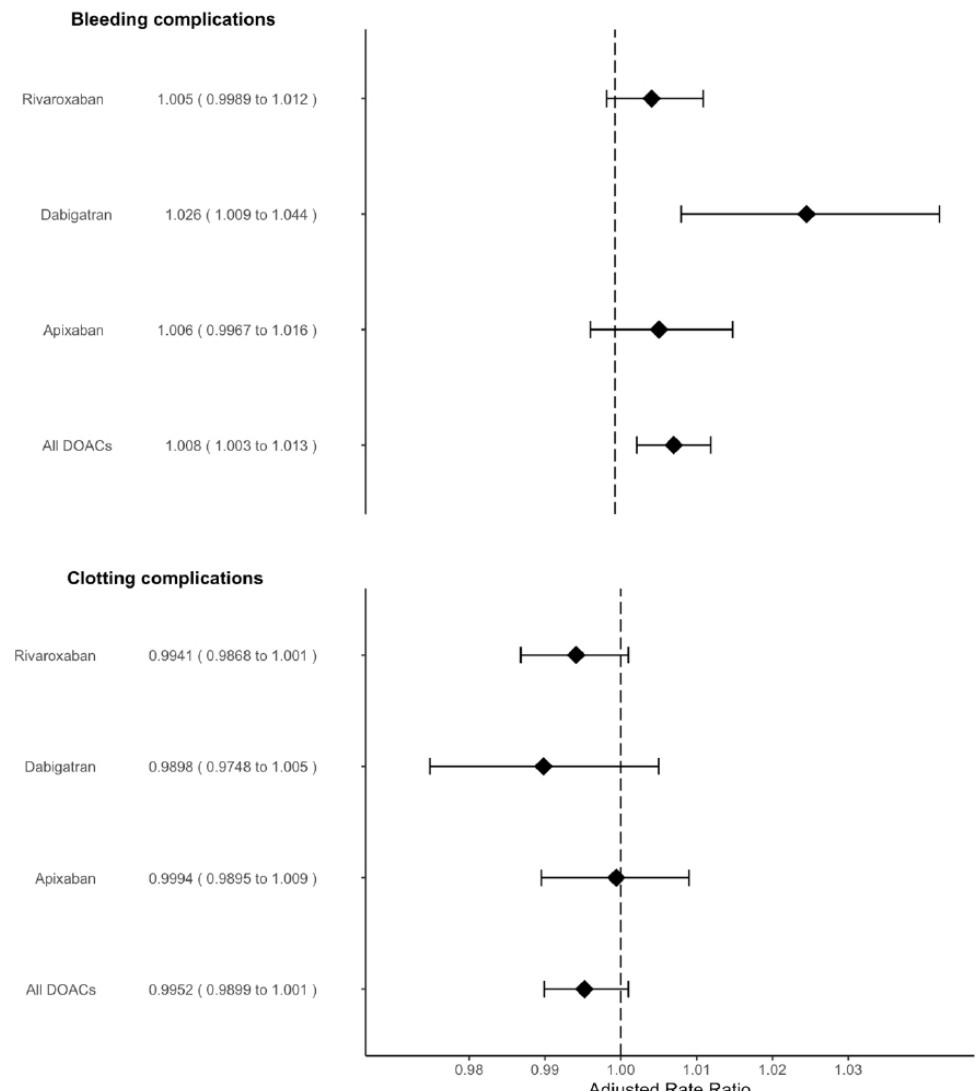

**Figure 3** Estimates from the regression model showing the relative change in the rate of emergency admissions for bleeding and clotting complications associated with each additional 10% of direct oral anticoagulants (DOACs) prescribed as a proportion of all anticoagulants.

The uptake of DOACs has not been consistent and there are geographical areas with a markedly higher rate of DOAC prescribing. Worryingly, the higher rates of DOAC prescribing were associated with the higher rates of hospital admissions for bleeding events, with a slight decline in emergency admissions for thromboembolic complications. We projected that with every 10% increase of DOAC prescribing as a proportion of all anticoagulants, there was a 0.8% increase in emergency admissions for bleeding complications. Moreover, we estimated from our regression model that between 2011 and 2016, there were an additional 4929 emergency admissions for bleeding complications, more than what would have been expected if DOACs had not been introduced. At the end of 2016, 94% of the total expenditure on anticoagulants prescribed by GP practices in England was spent on DOACs, which represented only 35% of all prescribed anticoagulants.

### Strengths and limitations of the study

The strength of this study is that we analysed data over several years that included the changes in the guidelines for prescribing anticoagulants. In 2014, the UK NICE guidelines recommended the use of DOACs and warfarin taking into consideration clinician and patient preferences. Comparable efficacy and the evidence from RCTs of better safety of DOACs compared with warfarin, combined with no need for regular monitoring and a wider therapeutic range, resulted in a major increase in prescribing.[1 3 4 6] We captured this increase in our longitudinal analyses. Our sample includes the majority of GP practices in England and is, therefore, able to estimate effects across the whole population. Our model assessed the association between the trend in prescribing and the trend in hospital admissions within each GP practice and we included a dummy variable for each year to account for the national trends. Any bias in our model of

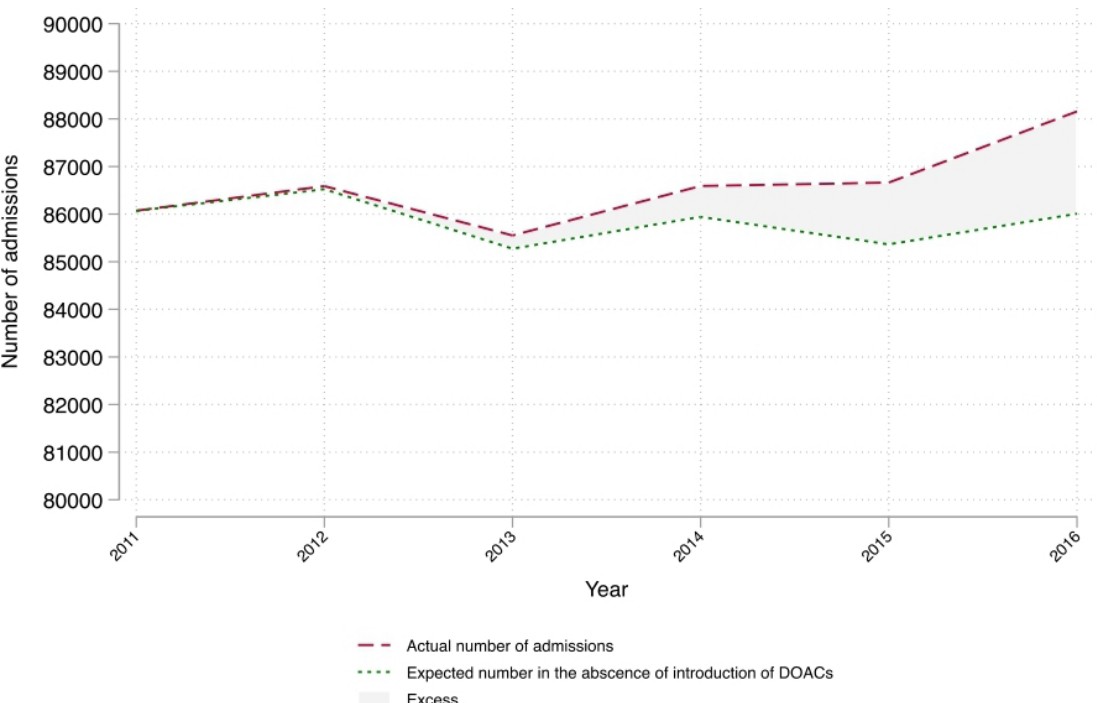

**Figure 4** Trend in emergency admissions for bleeding conditions between 2011 and 2016 and the trend estimated from the regression model that would have been expected if direct oral anticoagulants (DOACs) had not been introduced.

the association between our main exposure variable and outcome is therefore unlikely to be the result of time-invariant confounders that differ between GP practice population, or from any changes over time that affect all GP practices equally. We found a similar effect when using alternative models. When replicating the analysis with a common emergency admission we would not expect to be affected by DOAC prescribing such as gastrointestinal infections, we found, as expected, no association. The additional strength of our study is that we included in our analyses both, effectiveness and safety data on anti-coagulants used in England. We investigated whether the increased bleeding risk has been compensated by a reduction in risk from clotting conditions and found that although estimates of RRs were less than 1, CIs crossed 1, indicating that this result could have occurred even if there was no true association between DOAC prescribing and emergency admissions for clotting complications.

A number of limitations remain. First, the ecological design meant that it was not possible to link information on the prescribing for specific individuals to particular hospital admissions. Therefore, the associations observed at the aggregate level may not reflect associations at the level of the individual patients. Second, while our analysis adjusts for a number of observed and unobserved confounders, it is still possible that there are unobserved trends in factors that increase DOAC prescribing while also increasing the risk of bleeding complications. Bias could be introduced for example if there are differential trends between GP practices in the underlying risk of bleeding complications, that are not accounted for by our control variables and this increased risk of bleeding

complications led to increased DOAC prescribing in these practices. As our outcomes were measured at the population level, our study will, however, be less at risk of selection bias than studies using individual follow-up, where this results from clinical decisions to preferentially prescribe DOACs to patients with a higher risk of bleeding complications. Third, we could not control for adherence to medications in our study. Adherence to anticoagulants has been reviewed recently[14] with some contradictory reports. Although it is expected that adherence would be better for DOACs because of no need for regular moni-toring, regular INR tests required to ensure the correct dose of warfarin lead to stringent check-ups for adher-ence, while DOACs monitoring is not recommended and therefore adherence declines after the initial period.[15] With long-term adherence monitoring, our estimates would have been more precise.

### Comparisons with previous studies
Our results differ from the findings of several large-scale non-inferiority RCTs that have demonstrated better bleeding profiles of DOACs over warfarin.[5 16–21] This may not be surprising as trial data may provide limited information on relatively rare adverse effects due to low power and short follow-up. In addition, participants in clinical trials may be younger with a fewer comorbidities. For example, a renal impairment that occurs in the older population may be relevant. Renal clearance is more dominant for DOACs compared with warfarin, as 80%, 50%, 36% and 27% of unchanged dabigatran, edoxaban, rivaroxaban and apixaban, respectively, are excreted in the urine.[22] In addition, there is limited information

available on the risk-benefit profile of DOACs in patients with severe renal function impairment[23–26] particularly in patients with diverse ethnic backgrounds.[27] Recently, high inter-individual variability in DOAC plasma levels was observed in clinical practice.[28]

Our findings also differ from the results of a recent retrospective cohort study in patients with the intracranial haemorrhage who were previously prescribed warfarin or DOACs. Prior use of DOACs compared with prior use of warfarin was associated with a lower risk of hospital mortality.[29] A recent network meta-analysis by López-López also showed that DOACs are safer than warfarin in relation to major and intracranial bleeding.[6] However interestingly, in that meta-analysis, the risk of gastrointestinal bleeding was higher with dabigatran, edoxaban and rivaroxaban than with warfarin. In addition, edoxaban (30 mg and 60 mg two times per day) significantly increased the risk of clinically relevant bleeding compared with warfarin.[6] Although our study uses a longitudinal ecological study design, at least in part, our results are comparable with previous studies. In a large (>59 000 participants) population-based observational study conducted in Canada and the USA, the risk of hospital admissions for major bleeding or all-cause mortality in the first 90 days of treatment was similar for DOACs and warfarin.[30] Furthermore, in a study that used a US commercial database of 38 million people, warfarin users were hospitalised longer, stayed longer in an intensive care units than dabigatran or rivaroxaban users, but there was no difference in 30-day or 90-day all-cause mortality.[31]

It has been estimated that the cost of DOACs will rise sharply before patent expiry (in 2022) and by the year 2020 will constitute approximately 5% of the total NHS drug budget.[15] The drugs budget is the NHS's second biggest cost after its staff.[32] Cost of warfarin including INR monitoring for one patient per annum has been estimated to be £220 (https://www.gwh.nhs.uk/media/236108/doacs-for-dvt-pe-august-2016-v-9.pdf), while the cost of rivaroxaban (15 mg two times per day for 3/52 loading (provided by the hospital), then 20 mg one time a day) has been estimated to £657 p.a. (https://www.gwh.nhs.uk/media/236108/doacs-for-dvt-pe-august-2016-v-9.pdf). Similar costs have been estimated for other DOACs. Additionally, there is a huge discrepancy in the price of agents that are used to reverse the anticoagulant effects of warfarin (antidote vitamin K at £0.38) or DOACs (idarucizumab, a reversal agent for dabigatran at £2400 for a single treatment course of 2×2.5 g infusions (https://www.nice.org.uk/advice/esnm73) or andexanet alfa, a reversal agent for factor Xa inhibitors approved by the U.S. Food and Drug Administration (FDA) in May 2018, at approximately £1500). These high costs of DOACs could be avoided if warfarin treatment is optimised, for example through novel methods such as genotype-guided dosing[33] [34] and point-of-care INR monitoring,[35] and only individuals who are at the higher risk of developing bleeding events, for example those with variant alleles that increase

the risk of bleeding from warfarin,[36] could be prescribed DOACs. Genotype-guided dosing of warfarin has been shown to be cost-effective.[37] It has been shown previously that approximately 55% of variability in warfarin dose requirements can be estimated from clinical and genetic data of three polymorphisms in the warfarin molecular target (vitamin K epoxide reductase gene) and metabolising enzyme (CYP2C9 gene) and genotype-guided dosing can reduce the risk of major bleeding.[38 39]

Conclusions

This study is one of a few studies in the UK[1 40 41] that have evaluated the association between the rise in prescribing of DOACs after changes in NICE guidelines in 2014 and emergency hospital admissions for bleeding events. We found that the rate of emergency admissions for bleeding conditions increased to a greater extent in GP practices that were more likely to prescribe DOACs compared with warfarin. With rapidly increasing use of DOACs, these potential adverse consequences need to be taken into account when assessing the benefits and costs of anticoagulant treatment in clinical practice. It is not clear whether the DOACs are being prescribed without adequate notice being taken of restrictions and warnings in the summary of product characteristics, and in addition, whether closer monitoring is required in certain patient groups to increase the safety of use in clinical practice. The recently announced review by the EMA is thus important in order to further optimise the use of all oral anticoagulants so that risks are minimised while at the same time maximising benefits.

**Contributors** AA: substantial contributions to the conception and design of the work, drafting a manuscript. Agreed to be accountable for all aspects of the work ensuring that questions related to the accuracy or integrity of any part of the work are appropriately investigated and resolved. BB: substantial contribution to the design of the work, the acquisition, analysis and interpretation of data. Agreed to be accountable for all aspects of the work ensuring that questions related to the accuracy or integrity of any part of the work are appropriately investigated and resolved. MP: substantial contribution to the interpretation of the results, critical revision of the manuscript for important intellectual content and final approval of the manuscript .JD: critical revision of the manuscript for important intellectual content and final approval of the manuscript. KD: substantial contribution to data acquisition, data analysis and interpretation of data for the work. Critical revision of the manuscript for important intellectual content and final approval of the manuscript. TC: critical revision of the manuscript for important intellectual content and final approval of the manuscript. Preparation of plain language summary for dissemination of findings.

**Funding** This work was supported by the National Institute for Health Research, Collaboration for Leadership and Health Research and Care North West Coast (NIHR CLAHRC NWC) and the state Department of Health and Social Care.

**Disclaimer** The views expressed are those of the author(s) and not necessarily those of the NHS, the NIHR or the Department of Health.

**Map disclaimer** The depiction of boundaries on this map does not imply the expression of any opinion whatsoever on the part of BMJ (or any member of its group) concerning the legal status of any country, territory, jurisdiction or area or of its authorities. This map is provided without any warranty of any kind, either express or implied.

**Competing interests** We have read and understood BMJ policy on declaration of interests and declare the following: all authors have completed the ICMJE uniform disclosure form at http://www.icmje.org/coi_disclosure.pdf and declare: no support from any organisation for the submitted work (or describe if any); no financial relationships with any organisations that might have an interest in the submitted work in the previous 3 years (or describe if any); no other relationships

or activities that could appear to have influenced the submitted work (or describe if any).

**Patient and public involvement** Patients and/or the public were involved in the design, or conduct, or reporting, or dissemination plans of this research. Refer to the Methods section for further details.

**Patient consent for publication** Not required.

**Ethics approval** Ethical approval was not required for this ecological study data analyses.

**Provenance and peer review** Not commissioned; externally peer reviewed.

**Data availability statement** Data are available upon reasonable request. The dataset from this study will be made available on request to benbarr@liverpool.ac.uk.

**ORCID iD**
Ana Alfirevic http://orcid.org/0000-0002-2801-9817

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
