## [Reviewer comments · BMJ Open]

ARTICLE DETAILS

TITLE (PROVISIONAL)	Has the introduction of direct oral anticoagulants (DOACs) in England increased emergency admissions for bleeding conditions? A longitudinal ecological study.
AUTHORS	Alfirevic, Ana; Downing, Jennifer; Daras, Konstantinos; Comerford, Terence; Pirmohamed, Munir; Barr, Ben

VERSION 1 – REVIEW

REVIEWER	Jerrold Levy, MD Duke University School of Medicine Durham, NC, USA
REVIEW RETURNED	13-Sep-2019

GENERAL COMMENTS	The authors investigated if DOACs introduced in England compared with vitamin-K antagonists changed admissions for bleeding or thromboembolic complications by evaluating 5508 GP practices between 2011 and 2016 by evaluating the rate of emergency admissions to hospital in a longitudinal ecological study. They used panel regression models to examine the association between DOAC prescribing and emergency admission rates for bleeding and thromboembolic conditions. The found that for each additional 10% of DOACs prescribed as a proportion of all anticoagulants, there was a 0.7% increase in bleeding complications, and was associated with additional 4929 emergency admissions for bleeding complications. However, the increased DOAC prescribing was associated with a slight decline in admission for thromboembolic conditions. The authors suggest with the increasing use of DOACs, potential adverse consequences need to be taken into account when assessing the benefits and costs of anticoagulant treatment in clinical practice. Comments: 1. In reviewing your data, the bleeding complications seem to be more prevalent in the dabigatran treated patients. Because dabigatran is the most sensitive to renal dysfunction, and because practice patterns changed over the introduction of the different DOACs, it would be helpful to analyze this information if possible, in addition, vis-à-vis agents prescribed.2. How has DOAC use changed in your country over the course of approval and prescribing practices? Could that have influenced your results?3. In your discussion, I think it would be helpful to note that in the different clinical trials comparing DOACs to warfarin/VKAs, the use
--

	of VKAs were only consistently therapeutic ~ 65% of the time, and that is with frequent monitoring. The drug levels with the DOACs are more consistent and without monitoring. Please be sure to mention some of these differences. 4. Your report needs to have a section that describes the limitations of your report, the potential limitations of the data, and anything else that may have confounded your analysis and interpretation. What are the limitations of the NHS digital data that you used? Despite your thoughtful analyses, I think retrospective analyses of large databases are always fraught with issues. 5. Please comment whether prior analysis of the real-world experience from databases in the US have supported the authors' findings.
--	---

REVIEWER	Teresa Pérez Department of Statistics and Data Science, Complutense University of Madrid, Madrid, Spain
REVIEW RETURNED	15-Oct-2019

GENERAL COMMENTS	Thank you for the opportunity to review this manuscript. I have the following comments:  - A Poisson regression model was fitted including GP practice as a fixed factor. However, since the aim was to take into account the correlation within GP practice, and not to measure the effect size of each of them, a mixed model or a generalized estimating equation approach should be considered instead. -The explanatory variable “annual DOAC prescribing rate” was defined as the percentage of DOAC items prescribed in proportion to all anticoagulants items prescribed for each GP practice. In the analysis for dabigatran, apixaban and rivaroxoban, how did the authors define the explanatory variable? - The RR obtained was 1.007, and the authors interpreted “For each additional 10% of DOACs prescribed as a proportion of all anticoagulants, there was a 0.7% increase...” Review this interpretation, I think it should be rewritten as “For each additional 1% of DOACs prescribed as a proportion of all anticoagulants, there was a 0.7% increase...” - When fitting a Poisson regression model, the RR is referred as rate ratio, not risk ratio. Change the label in the x-axis in figures. - In this sentence “Overall the introduction of DOACs between 2011 and 2016 was associated with an additional 4929 emergency admissions for bleeding complications” Suggest give this information also as a percentage of the population size.
--

VERSION 1 – AUTHOR RESPONSE

Reviewer(s)' Comments to Author:

- > Reviewer: 1
- > Reviewer Name: Jerrold Levy, MD
- >
- > Institution and Country: Duke University School of Medicine Durham, NC, USA
- >
- > Please state any competing interests or state 'None declared': none

>

> Please leave your comments for the authors below The authors investigated if DOACs introduced in England compared with vitamin-K antagonists changed admissions for bleeding or thromboembolic complications by evaluating 5508 GP practices between 2011 and 2016 by evaluating the rate of emergency admissions to hospital in a longitudinal ecological study. They used panel regression models to examine the association between DOAC prescribing and emergency admission rates for bleeding and thromboembolic conditions. They found that for each additional 10% of DOACs prescribed as a proportion of all anticoagulants, there was a 0.7% increase in bleeding complications, and was associated with additional 4929 emergency admissions for bleeding complications. However, the increased DOAC prescribing was associated with a slight decline in admission for thromboembolic conditions. The authors suggest with the increasing use of DOACs, potential adverse consequences need to be taken into account when assessing the benefits and costs of anticoagulant treatment in clinical practice.

>

> Comments:

>

> 1. In reviewing your data, the bleeding complications seem to be more prevalent in the dabigatran treated patients. Because dabigatran is the most sensitive to renal dysfunction, and because practice patterns changed over the introduction of the different DOACs, it would be helpful to analyze this information if possible, in addition, vis-à-vis agents prescribed.

Response:

We thank the reviewer for the comment. Our analyses were repeated for individual DOACs dabigatran, rivaroxaban and apixaban. We mentioned these analyses on page 7 of the manuscript (line 7-12). We have also included our individual DOACs results in our new Figure 3.

> 2. How has DOAC use changed in your country over the course of approval and prescribing practices? Could that have influenced your results?

Response:

We have clarified now in our figure 1 and 2 legends that the NICE guidance changes in 2014 have contributed to increased use of DOACs.

> 3. In your discussion, I think it would be helpful to note that in the different clinical trials comparing DOACs to warfarin/VKAs, the use of VKAs were only consistently therapeutic ~ 65% of the time, and that is with frequent monitoring. The drug levels with the DOACs are more consistent and without monitoring. Please be sure to mention some of these differences.

Response:

In Phase III clinical trials comparing DOACs to warfarin fixed-dose administration was proposed for DOACs, without the necessity for routine laboratory monitoring. However, in clinical practice a high inter-individual variability in DOAC plasma levels was observed (Testa et al., Plasma levels of direct oral anticoagulants in real life patients with atrial fibrillation: Results observed in four anticoagulation clinics, Thrombosis Research 2016:178-183.). Furthermore, in patients with low plasma levels of DOACs thromboembolic complications have been identified (Testa et al. Low drug levels and thrombotic complications in high-risk atrial fibrillation patients treated with direct oral anticoagulants Journal of Thrombosis and Haemostasis 2018;16(5):842-848). We have now updated information on pages 4 and 11 of the manuscript by adding the following sentences: 1) Its (warfarin) use is associated with large inter-individual variability in dose requirements, narrow therapeutic index, slow onset of action, low time in therapeutic range and concomitant drug and food interactions. 2) In addition, in clinical trials fixed-dose administration was proposed for DOACs, without the need for routine laboratory monitoring. 3) Recently, high inter-individual variability in DOAC plasma levels was observed in clinical practice

> 4. Your report needs to have a section that describes the limitations of your report, the potential limitations of the data, and anything else that may have confounded your analysis and interpretation.

What are the limitations of the NHS digital data that you used? Despite your thoughtful analyses, I think retrospective analyses of large databases are always fraught with issues.

Response:

Please see previously response to one of the editor's comments. We have highlighted in the discussion part of the manuscript where we mention limitations of the study, they are described in some detail on page 10.

> 5. Please comment whether prior analysis of the real-world experience from databases in the US have supported the authors' findings.

Response:

We would like to thank the reviewer for this comment. We found information on real-life study in the US by Charlton et al. 2018 that compared the length of hospitalisation and mortality for bleeding with anticoagulants. The authors found that bleeding with the newer agents, rivaroxaban and dabigatran, was not more dangerous than bleeding associated with warfarin. However, the authors did not investigate the risk of bleeding in this retrospective study. We have included a sentence to the manuscript:

Furthermore, in a study that used an US commercial database of 38 million people warfarin users were hospitalised longer, stayed longer in an intensive care unit than dabigatran or rivaroxaban users, but there was no difference in 30- or 90-day all-cause mortality (Charlton, 2018).

> Reviewer: 2

> Reviewer Name: Teresa Pérez

>

> Institution and Country: Department of Statistics and Data Science,

> Complutense University of Madrid, Madrid, Spain

>

> Please state any competing interests or state 'None declared': None

> declared

>

> Please leave your comments for the authors below Thank you for the

> opportunity to review this manuscript. I have the following comments:

> - A Poisson regression model was fitted including GP practice as a fixed factor. However, since the aim was to take into account the correlation within GP practice, and not to measure the effect size of each of them, a mixed model or a generalized estimating equation approach should be considered instead.

Response:

We thank the reviewer for their comment and agree it is important to ensure the longitudinal nature of the data is accounted for in the analysis. Firstly, we include a fixed effect for each GP practice to remove potential time-invariant differences between GP practices that could be confounders in the analysis [1]. The purpose is not to measure the effect of each GP practice. We describe this on Page 6 lines 39-54. Whilst this approach to addressing problems of unobserved confounding is less frequently used in epidemiological studies, it is applied more in economics Gunasekera et al 2013 provides a full explanation of the approach. [1] A mixed effects model - for example a model with a random intercept for each GP practice could introduce bias as this assumes that the effect of each GP practice is uncorrelated with other variables in the model, and this is unlikely to be true.[2] The fixed effects method we use is therefore more conservative and less likely to be subject to bias than that suggested by the reviewer.

A second issue we need to account for is heteroskedasticity, resulting from the variance within GP practices (over time) being smaller than the variance between practice. We do this by using Huber- White clustered standard errors that are robust to divergence of the data from the assumptions of a Poisson distribution and the clustering of variance between GP practices, as mentioned on page 7 lines 3-7.[3] This is the same as using a Generalised Estimating Equation and assuming an independent correlation structure (see [4].) However to explore whether our findings are sensitive to alternative model assumptions we have additionally added results to appendix 3 (1) Using a general estimating equation, with an exchangeable correlation structure and a general estimating equation, with an autoregressive correlation structure (AR1). With both of these models we find very similar effects.

- 1 Gunasekara FI, Richardson K, Carter K, et al. Fixed effects analysis of repeated measures data. *Int J Epidemiol* 2013;:dyt221. doi:10.1093/ije/dyt221***
- 2 Clarke P, Crawford C, Steele F, et al. The Choice between Fixed and Random Effects Models: Some Considerations for Educational Research. *Institute for the Study of Labor (IZA) Discussion Papers* 2010;5287:36.***
- 3 Rogers W. Regression standard errors in clustered samples. *Stata Technical Bulletin* 1994;3.<http://ideas.repec.org/a/tsj/stbull/y1994v3i13sg17.html> (accessed 3 Oct 2013).***
- 4 Generalized estimating equations | Stata. <https://www.stata.com/features/generalized-estimating-equations/> (accessed 18 Nov 2019).***

We also made changes to Appendices 3 and 4.

3. General Estimating equation - with exchangeable correlation structure.

Estimates from regression model showing the relative change in the rate of emergency admissions for bleeding and clotting complications associated with each additional 10% of DOACs prescribed as a proportion of all anticoagulants

4. General Estimating equation - autoregressive(AR-1) correlation structure. Estimates from regression model showing the relative change in the rate of emergency admissions for bleeding and clotting complications associated with each additional 10% of DOACs prescribed as a proportion of all anticoagulants

> -The explanatory variable “annual DOAC prescribing rate” was defined as the percentage of DOAC items prescribed in proportion to all anticoagulants items prescribed for each GP practice. In the analysis for dabigatran, apixaban and rivaroxaban, how did the authors define the explanatory variable?

Response:

We have added this sentence to the manuscript: The prescribing rate for each separate DOAC (dabigatran etexilate, apixaban, edoxaban or rivaroxaban) was calculated as the percentage of items of these medications prescribed as a proportion of all anticoagulants items prescribed.

> - The RR obtained was 1.007, and the authors interpreted “For each additional 10% of DOACs prescribed as a proportion of all anticoagulants, there was a 0.7% increase...” Review this

interpretation, I think it should be rewritten as “For each additional 1% of DOACs prescribed as a proportion of all anticoagulants, there was a 0.7% increase...”

Response:

We thank the reviewer for this comment. What we have said in the manuscript is correct. As the effect for each additional 1% of DOACs prescribed is very small (a 0.07% increase), for ease of interpretation we present the results for a 10% increase, which is a meaningful level of increase given that the average prescribing rate increased from 0 to 35% during this time. We have added a line to the text on page 8. to clarify this.

> - When fitting a Poisson regression model, the RR is referred as rate ratio, not risk ratio. Change the label in the x-axis in figures.

Response:

Thank you, we have changed the figures as suggested.

New Figure 3

> - In this sentence “Overall the introduction of DOACs between 2011 and 2016 was associated with an additional 4929 emergency admissions for bleeding complications” Suggest give this information also as a percentage of the population size.

Response:

We have included the following statement:

Overall the introduction of DOACs between 2011 and 2016 was associated with 1.94 additional emergency admissions for bleeding complications per 100,000 people per year (0.98 to 2.9), the equivalent of 4929 emergency admissions for bleeding complications.

VERSION 2 – REVIEW

REVIEWER	Jerrold H Levy, MD Duke University Medical Centre, Durham, NC
REVIEW RETURNED	26-Nov-2019

GENERAL COMMENTS	The authors have thoughtfully responded to my comments. I have no further questions.
--

REVIEWER	Teresa Pérez Complutense University of Madrid, Statistics and Data Science
REVIEW RETURNED	06-Dec-2019

GENERAL COMMENTS	Thank you for your answers, however I still have some concerns related to your manuscript. 1.- Yes, I agree with you that the fixed effects method is becoming quite popular, particularly on the big data analysis. In that context, the massive sample size and high dimensionality within each cluster, makes this approach maybe the best choice. However, the situation is completely different in your study. You are trying to solve three problems: to adjust by GP practice (solved by including GP in the model as fixed factor); to take the within GP practice correlation into account (solved considering Huber-White clustered standard error) and to study the over-dispersion (solved fitting a negative binomial model as a sensitivity analysis). These three issues can be solved directly by using the generalized estimating equations method. In addition this approach has several advantages:  1. - Although it is a robust method, the closer the working matrix is to the true structure, the more efficient the estimates will be. 2. - Correlation is allowed to differ by year. 3.- It captures the over-dispersion. 4.- The relationship among the outcome variable and predictor variables is estimated using all available data, including data from GP practices with missing observations. 5.- It reduces the number of parameters to be estimated, from 5507 (number of dummies, in your case) to 21 (in the worst case, when the unstructured correlation structure is selected) or to 7 (in the best case, when the AR(1) correlation structure is selected). The covariance matrix is a 6x6 symmetric matrix, because there are 6 repeated measures collected from 2011 to 2016. 2.- I think you misunderstood this comment: The RR obtained was 1.007, and the authors interpreted “For each additional 10% of DOACs prescribed as a proportion of all anticoagulants, there was a 0.7% increase...” Review this interpretation, I think it should be rewritten as “For each additional 1% of DOACs prescribed as a proportion of all anticoagulants, there was a 0.7% increase...”
--

	The interpretation of RR is that increasing one unit the predictor variable (in this case each additional 1% of DOACs prescribed as a proportion of all anticoagulants) increases the rate by 0.7%, $(RR-1)*100$, unless something else is mentioned (e.g. standardized coefficients). So when it increases 10 units (in this case increasing a 10% of DOACs prescribed as a proportion of all anticoagulants) the new RR is $(1.007)**10=1.07$, which means that the rate increases by 7% (not 0.7%). 3.- . Please present the results in a statistically manner avoiding references to “was associated” when this association has not been proven, e.g. “Increased use of DOACs was associated with a small decline in emergency admissions for clotting related conditions; however this was not statistically significant ($p=0.065$), see Figure 3.”
--	--

VERSION 2 – AUTHOR RESPONSE

Our responses to the reviewer’s comments:

Thank you for your answers, however I still have some concerns related to your manuscript.

1.- Yes, I agree with you that the fixed effects method is becoming quite popular, particularly on the big data analysis. In that context, the massive sample size and high dimensionality within each cluster, makes this approach maybe the best choice. However, the situation is completely different in your study. You are trying to solve three problems: to adjust by GP practice (solved by including GP in the model as fixed factor); to take the within GP practice correlation into account (solved considering Huber-White clustered standard error) and to study the over-dispersion (solved fitting a negative binomial model as a sensitivity analysis). These three issues can be solved directly by using the generalized estimating equations method. In addition this approach has several advantages:

1. - Although it is a robust method, the closer the working matrix is to the true structure, the more efficient the estimates will be.
2. - Correlation is allowed to differ by year.
- 3.- It captures the over-dispersion.
- 4.- The relationship among the outcome variable and predictor variables is estimated using all available data, including data from GP practices with missing observations.
- 5.- It reduces the number of parameters to be estimated, from 5507 (number of dummies, in your case) to 21 (in the worst case, when the unstructured correlation structure is selected) or to 7 (in the best case, when the AR(1) correlation structure is selected). The covariance matrix is a 6x6 symmetric matrix, because there are 6 repeated measures collected from 2011 to 2016.

Response.

We thank the reviewer for these comments. We have already provided sensitivity analysis using the model as proposed by the reviewer – i.e using generalized estimating equations methods, showing that this doesn't make any substantial difference to our results. Please see appendix 3. This clearly indicates that these differences in modelling make little difference and are therefore not a source of bias. As the reviewer indicates our analysis as originally planned addresses the three issues raised. Although as the reviewer points our original analysis may be less efficient and therefore more conservative, we have kept our main analysis as that originally laid out in our analysis plans and present the alternative proposed by the reviewer in an appendix. This would seem to be a reasonable compromise between addressing the issues raised by the reviewer, whilst being consistent with our pre-specified analysis plans.

2.- I think you misunderstood this comment:

The RR obtained was 1.007, and the authors interpreted “For each additional 10% of DOACs prescribed as a proportion of all anticoagulants, there was a 0.7% increase...” Review this interpretation, I think it should be rewritten as “For each additional 1% of DOACs prescribed as a proportion of all anticoagulants, there was a 0.7% increase...”

The interpretation of RR is that increasing one unit the predictor variable (in this case each additional 1% of DOACs prescribed as a proportion of all anticoagulants) increases the rate by 0.7%, $(RR-1)*100$, unless something else is mentioned (e.g. standardized coefficients). So when it increases 10 units (in this case increasing a 10% of DOACs prescribed as a proportion of all anticoagulants) the new RR is $(1.007)^{10}=1.07$, which means that the rate increases by 7% (not 0.7%).

Response.

The RR are per 10 percentage point increases – because we have scaled them as such – as we state “We present the effect estimates per additional 10% of DOACs prescribed as this reflects a more meaningful level of increase than a 1 percentage point increase.” The rate ratio for a 1 percent increase is exactly the same but with an additional zero after the decimal point i.e RR:1.0007 95% CI 1.0002 to 1.0013. We have rescaled this to give it per 10 percentage points increase i.e we have divided these by 10 to give the effect per 10 percentage points. We have added further text to explain this.

We did however also notice that there was an inconsistency between the results in the text and those in Figure 3, this was because the results from Appendix 3 model 1 had been entered in the text in error. We have changed this in the main text.

3.- . Please present the results in a statistically manner avoiding references to “was associated” when this association has not been proven, e.g. “Increased use of DOACs was associated with a small decline in emergency admissions for clotting related conditions; however this was not statistically significant ($p=0.065$), see Figure 3.”

Response.

We have reworded the results to present them as observed and provide an explanation of the meaning of the p value. We have aimed to follow the American Statistical Associations guidance on p-values that they do not indicate whether a particular association is proven or not and that “a conclusion does not immediately become “true” on one side of a divide [$p<0.05$] and “false” on the other.”[1]

VERSION 3 – REVIEW

REVIEWER	Teresa Pérez Department of Statistics and Data Sciences. Complutense University of Madrid. Spain
REVIEW RETURNED	20-Jan-2020

GENERAL COMMENTS	1.- I am not entirely convinced with the authors’ argument. From a statistical point of view, the aim it is not about obtaining similar results with both methods, it is about using the best approach as a guide for future researchers. 2.- I have to say that the last part of this sentence is not true: “We present the effect estimates (Rate Ratios) per additional 10% of DOACs prescribed, rather than for each additional 1 percentage point increase as this reflects a more meaningful level of increase than a 1 percentage point increase. This is achieved by dividing the rate ratio by 10 to give the effect per 10 percentage points of DOAC prescribing” Indeed, it is achieved by transforming the rate ratio to the power of 10, in this case $1.0007^{10}=1.007$, but I don’t think it is necessary to include it. 3.- The aim of the American Statistical Associations guidance and the rest of publications in this line, is to avoid reference to testing procedures and strong p-value statements, in favour of the size effect (which should be determined before analysing the data). So I suggest not mentioning anything about the p-value and just focusing on the rate ratio and its confidence interval. 4.- Please check the confidence intervals in Appendix 3, Figure 4, (1.009 to 1.009)
---

VERSION 3 – AUTHOR RESPONSE

1.- I am not entirely convinced with the authors' argument.

From a statistical point of view, the aim it is not about obtaining similar results with both methods, it is about using the best approach as a guide for future researchers.

Response : We have replaced our main analysis with the GEE analysis and put our original analysis in the appendix.

2.- I have to say that the last part of this sentence is not true:

“We present the effect estimates (Rate Ratios) per additional 10% of DOACs prescribed, rather than for each additional 1 percentage point increase as this reflects a more meaningful level of increase than a 1 percentage point increase. This is achieved by dividing the rate ratio by 10 to give the effect per 10 percentage points of DOAC prescribing“

Indeed, it is achieved by transforming the rate ratio to the power of 10, in this case

$1.0007^{10}=1.007$, but I don't think it is necessary to include it.

Response: Thanks – of course the reviewer is correct, the transformation was described in error as dividing by 10 rather than transforming the rate ratio to the power of 10 - i.e $1.0007^{10}=1.007$, as the reviewer has said it is probably not necessary to include this, so we have removed this sentence.

3.- The aim of the American Statistical Associations guidance and the rest of publications in this line, is to avoid reference to testing procedures and strong p-value statements, in favour of the size effect (which should be determined before analysing the data). So I suggest not mentioning anything about the p-value and just focusing on the rate ratio and its confidence interval.

Response: Thanks – we have changed this to just refer to confidence intervals.

4.- Please check the confidence intervals in Appendix 3, Figure 4, (1.009 to 1.009)

Response: Thanks – these have been corrected.